# LABEL-DISTRIBUTION-AGNOSTIC ENSEMBLE LEARNING ON FEDERATED LONG-TAILED DATA

## ABSTRACT

Federated Learning (FL) is a distributed machine learning paradigm that enables devices to collaboratively train a shared model. However, the long-tailed distribution in nature deteriorates the performance of the global model, which is difficult to address due to data heterogeneity, *e.g.*, local clients may exhibit diverse imbalanced class distributions. Moreover, existing re-balance strategies generally utilize label distribution as the class prior, which may conflict with the privacy requirement of FL. To this end, we propose a Label-Distribution-Agnostic Ensemble (LDAE) learning framework to integrate heterogeneous data distributions using multiple experts, which targets to optimize a balanced global objective under privacy protection. In particular, we derive a privacy-preserving proxy from the model updates of clients to guide the grouping and updating of multiple experts. Knowledge from clients can be aggregated via implicit interactions among different expert groups. We theoretically and experimentally demonstrate that (1) there is a global objective gap between global and local re-balance strategies[1] and (2) with protecting data privacy, the proxy can be used as an alternative to label distribution for existing class prior based re-balance strategies. Extensive experiments on long-tailed decentralized datasets demonstrate the effectiveness of our method, showing superior performance to state-of-the-art methods.

## 1 INTRODUCTION

Federated Learning (FL) aims to collaboratively learn from data dominated by a number of remote clients and produce a highly accurate global model on the server with aggregated knowledge. The most important issues in practical FL applications mainly involve data heterogeneity and privacy protection during collaboration of disparate data sources. Such issues are even more significant in the setting of long-tailed data distribution for some real-world scenarios (Cui et al., 2019; Liu et al., 2019), such as medical applications (Li et al., 2019; Malekzadeh et al., 2021) and autonomous vehicles (Samarakoon et al., 2019; Pokhrel & Choi, 2020).

Under the long-tailed global data distribution, it is extremely challenging to learn an effective global model by leveraging knowledge from local clients. From the local perspective, there can be a large divergence among the imbalanced label distributions of different clients, resulting in the heterogeneous imbalance as shown in Figure 1(a), *i.e.*, local datasets on different clients may have different imbalance ratios or minority classes. From the global perspective, one should handle the imbalance issue with privacy preservation (Li et al., 2021a), *i.e.*, the server should not require clients to upload label distributions for re-balance strategies.

Several techniques have been proposed to tackle the class imbalance problem in FL, such as loss re-weighting (Wang et al., 2021; Shen et al., 2021), client clustering (Duan et al., 2020) and the client selection scheme (Yang et al., 2021). Most of them focus on datasets with only a few classes (*e.g.*, ten or twenty classes), suffering from significant performance drops on large-scale imbalanced datasets with more classes (Liu et al., 2019; Zhang et al., 2021b). Simultaneously, existing solutions generally assume that some sensitive information is accessible to the global server, *e.g.*, a balanced

---

[1]The local re-balance strategy means that each client utilizes re-balance methods based on the local label distribution, while the global re-balance strategy applies re-balance methods using global label distribution as the class-wise prior.

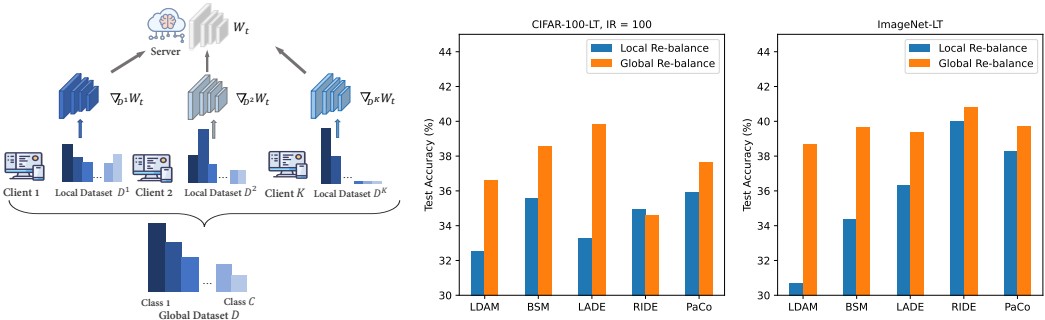

(a) Heterogeneous Imbalance in FL  (b) Global Re-balance vs. Local Re-balance

Figure 1: Illustration of the long-tailed FL problem. (a) Different local datasets exhibit diverse imbalanced label distributions, which even differ from the global dataset. (b) For existing class prior based re-balance algorithms, the global re-balance strategy outperforms the local one, motivating us to explore global prior information for the long-tailed FL problem.

mini-dataset (Duan et al., 2020; Wang et al., 2021) or learnable hyper-parameters of local clients (Shen et al., 2021), which may pose data privacy concerns for the realistic FL applications. Furthermore, we notice that the effectiveness of existing class prior based strategies (Cao et al., 2019; Ren et al., 2020; Hong et al., 2021) for the imbalance issue in FL remains under-explored.

In this work, we investigate the effectiveness of existing class prior based re-balance algorithms from the global and local perspectives under the long-tailed distribution. Experimentally, as indicated by Figure 1(b)), combined with these algorithms, the global re-balance strategy yields higher recognition accuracy than the local re-balance strategy in the setting of FL. We theoretically demonstrate that the main reason arises from the gap of objective functions between the global and local re-balance strategies in FL, where the former can yield the matched objective of the centralized training. However, obtaining global label distribution requires clients to upload their own label distributions to the server, which may violate the privacy protection principle in FL (Wang et al., 2021; McMahan et al., 2017). Thus, it is a critical issue to exploit privacy-preserving priors for global re-balance strategy to maintain a balanced global objective function.

To overcome the above-mentioned problem, we propose a **L**abel-**D**istribution-**A**gnostic **E**nsemble learning framework (LDAE) to deal with the data heterogeneity and privacy in the long-tailed FL setting. Specifically, we present the proxy information as the class prior of global re-balance strategies rather than label distribution. The proxy information is derived from the model updates uploaded by local clients, which is agnostic about the local label distributions for privacy protection. To alleviate the heterogeneous issue, we propose a multi-expert model architecture to aggregate the knowledge from different client groups, where clients in the same group have similar local data distribution and train a corresponding expert. The heterogeneity could be mitigated through information interaction among different experts trained on different local data distributions.

In conclusion, the key contributions of this work are:

(1) We experimentally and theoretically explore the effectiveness of existing class prior based re-balance algorithms in FL. It is demonstrated that there is a mismatch of objectives between local and global re-balance strategies, which indicates that the global re-balance performs better than the local one on the imbalanced decentralized data.

(2) To address the imbalance issue with privacy protection, we propose a novel FL framework called LDAE to utilize uploaded model updates to cluster clients into different groups, where a multi-expert architecture is used to aggregate the knowledge from different groups with heterogeneous data distribution. Our method is agnostic to the label distribution of the clients.

(3) The experimental results on multiple benchmark datasets demonstrate that LDAE can significantly outperform previous state-of-the-art (SOTA) methods under heterogeneous data distribution, simultaneously protecting the data privacy of the clients.

## 2 RELATED WORK

**Federated Learning.** FL (McMahan et al., 2017) is a learning framework to protect the data privacy of participants. There is a central server and multiple clients in FL. A global model is learned on distributed data from different clients. One of the most important challenges in FL is data heterogeneity. Many previous studies concentrated on this problem (Smith et al., 2017; Zhao et al., 2018; Sattler et al., 2020; Karimireddy et al., 2020; Li et al., 2020b; T Dinh et al., 2020). However, most of them consider that the global dataset (the dataset where all local data is centralized together) is balanced. Recently, some works (Duan et al., 2020; Yang et al., 2021; Wang et al., 2021; Shen et al., 2021) focused on the impact of class imbalance in FL and propose different strategies. However, these methods all require extra information other than model parameters on the server and may not be satisfied with the data privacy concern. Moreover, they only conduct experiments on datasets with a few classes, and their algorithms may not be efficient enough to deal with the large-scale long-tailed imbalance problem. CReFF (Shang et al., 2022) considers long-tailed data in FL inspired by (Kang et al., 2019). However, while claiming it has no privacy concerns, their work needs feature gradients of clients' data, which can be used to recover original data with model updates.

**Long-tailed Learning.** In real-world scenarios, data often has a long-tailed label distribution, where the majority classes have massive samples while the minority classes only have a few samples (Zhang et al., 2021b). There are many re-balance strategies proposed from different perspectives in long-tailed learning. Class re-sampling (Chawla et al., 2002; He & Garcia, 2009; Kang et al., 2019) is a common type, such as over-sampling the minority classes (Shen et al., 2016; Kang et al., 2019) or under-sampling (He & Garcia, 2009) the majority classes. Another scheme to learn a balanced model is loss re-weighting (Cost-sensitive Learning) (Sun et al., 2007; Cui et al., 2019; Lin et al., 2014). Generally, these methods tend to give a large training loss when the sample belongs to the minority class. Recently, many works in long-tailed learning have focused on learning a good representation extractor to improve the generalization ability of the model. PaCo (Cui et al., 2021) introduces the contrastive learning method to the long-tailed dataset. Ensemble learning based methods are also becoming more important in long-tailed learning (Zhou et al., 2020; Xiang et al., 2020; Wang et al., 2020; Zhang et al., 2021a; Cai et al., 2021). Until now, many re-balance strategies have worked well to solve the long-tail imbalance problem on centralized datasets. However, it remains a question whether they are useful in FL. In this work, we answer this question through theoretical analysis and experiments results. Then, we propose a novel algorithm to help these re-balance strategies maintain their effectiveness in FL in the following section.

## 3 METHOD

In this section, we first highlight the challenges when data heterogeneity encounters long-tailed distribution in FL. Then we systematically explore the local and global re-balance strategies as the motivation. Finally, we describe the proposed LDAE framework in detail.

**Notations.** We discuss a typical FL setting with $K$ clients indexed by $[K]$ and a central server. Each client $k$ has a local training dataset $\mathcal{D}^k$ with a total of $n^k$ samples. We call the dataset $\mathcal{D} = \bigcup_{k \in [K]} \mathcal{D}^k$ as the global dataset. Considering a $C$-class classification task on the global dataset $\mathcal{D}$, each class is indexed by $[C]$ and $(\boldsymbol{x}, y) \in \mathcal{X} \times [C]$ denotes a sample in $\mathcal{D}$, where $\boldsymbol{x}$ is an image in the input space $\mathcal{X}$ and $y$ is its corresponding label. Let $\theta \in \Theta$ be the model parameters and $f_\theta(\boldsymbol{x}, y)$ denote the loss of sample pair $(\boldsymbol{x}, y)$. We set $n_j$ as the number of training sample for class $j$, and $n = \sum_{j=1}^{C} n_j$ is the total number of training samples. Suppose $\mathcal{D}$ follows a long-tailed distribution, *i.e.*, the sample size is exponentially distributed *w.r.t.* class index. Without loss of generality, we assume that the classes are sorted by cardinality in decreasing order, *i.e.*, if $i < j$, then $n_i \geq n_j$. The global imbalance ratio is defined as $n_{max}/n_{min}$.

Typically, FL aims at learning a single shared model and optimizing the global objective, which is the aggregation of the local objective:

$$\min_{\theta \in \mathbb{R}^d} \sum_{k=1}^{K} \frac{n_k}{n} F_k(\theta), \quad where \quad F_k(\theta) = \frac{1}{n_k} \sum_{(\boldsymbol{x}, y) \in \mathcal{D}^k} f_\theta(\boldsymbol{x}, y). \tag{1}$$

However, the above formulation suffers from long-tailed data distribution and performs poorly on the minority classes. In a decentralized system with global imbalanced data distribution, the training data on a given client is typically based on the particular data source. Hence, local datasets may exhibit varying imbalanced label distributions and even differ from the global distribution. We refer to such circumstances as the following heterogeneous imbalance.

**Definition 1** *(Heterogeneous Imbalance). Given a global long-tailed training dataset for FL, we define heterogeneous imbalance by the following items: (1) Various imbalanced ratios or majority (minority) classes among local datasets. (2) Different imbalanced ratios or majority (minority) classes between local and global datasets.*

The definition reflects that the class imbalance issue is aggravated by data heterogeneity due to various local imbalanced distributions *w.r.t* the global one. As a theoretical motivation, we consider the following configuration of heterogeneous imbalance to investigate the effectiveness of prior based re-balance techniques for the class imbalance issue in FL. The theoretical results inspire us to optimize a balanced objective from a global perspective instead of a local one. We also verify the theoretical results via extensive experiments.

## 3.1 THEORETICAL MOTIVATION

We consider a binary classification problem where the ground truth is either positive ($y^+$) or negative ($y^-$). Under the imbalanced setting, we assume that two clients $C_0$ and $C_1$ have heterogeneously imbalanced data distributions, *i.e.*, client $C_0$ accesses $n_0^+$ positives and $n_0^-$ negatives, while client $C_1$ accesses $n_1^+$ positives and $n_1^-$ negatives. Without loss of generality, we consider a simple loss re-weighting strategy that takes the inverse of the proportion of each class as the weight of the loss for this class. To be specific, we set the weight of loss to be $n_0/n_0^+$ for the positive class and $n_0/n_0^-$ for the negative class on client $C_0$. Similarly, client $C_1$ uses $n_1/n_1^+$ for the positive class and $n_1/n_1^-$ for the negative class. For the global re-balance strategy, inverse of the global label distribution is taken as the weights of loss, (*i.e.*, $(n_0 + n_1)/(n_0^+ + n_1^+)$ for positive class and $(n_0 + n_1)/(n_0^- + n_1^-)$ for negative class). We denote the global objective of the global re-balance strategy and the local re-balance strategy as $G_g(\theta)$ and $G_l(\theta)$, respectively. Then we can measure the difference of global object functions between local and global re-balance strategies.

**Lemma 1** *Using the global re-balance strategy, the global objective yields the same form as the objective on the centralized dataset with re-balance methods:*

$$G_g(\theta) = \frac{1}{n_0^+ + n_1^+} \sum_{(\boldsymbol{x}, y^+) \in \mathcal{D}} f_\theta(\boldsymbol{x}, y^+) + \frac{1}{n_0^- + n_1^-} \sum_{(\boldsymbol{x}, y^-) \in \mathcal{D}} f_\theta(\boldsymbol{x}, y^-). \tag{2}$$

**Theorem 1** *Under the above setting, let $\mathcal{E}$ be the biased estimation of global label distribution. Then there exists a group of re-balance weights $e$ derived from $\mathcal{E}$, whose global objective $G_e$ satisfies:*

$$G_g(\theta) \le G_e(\theta) < G_l(\theta), \tag{3}$$

*where the objective gap $\Delta = G_l(\theta) - G_g(\theta)$ can be written as:*

$$\Delta = \frac{n_1(n_0^+)^2 + n_0(n_1^+)^2}{n_0^+ n_1^+ (n_0 + n_1)(n_0^+ + n_1^+)} \sum_{(\boldsymbol{x}, y^+) \in \mathcal{D}} f_\theta(\boldsymbol{x}, y^+) + \frac{n_1(n_0^-)^2 + n_0(n_1^-)^2}{n_0^- n_1^- (n_0 + n_1)(n_0^- + n_1^-)} \sum_{(\boldsymbol{x}, y^-) \in \mathcal{D}} f_\theta(\boldsymbol{x}, y^-). \tag{4}$$

**Interpretation.** The above analysis illustrates the following points: (1) The global re-balance strategy for the federated long-tailed problem optimizes the same objective function as the re-balance strategy on the centralized dataset. (2) The local re-balance strategy with a larger objective is less effective than the global one with a smaller objective. (3) There exists a re-balance strategy derived from the global perspective, which yields a smaller objective than the local re-balance strategy.

## 3.2 GLOBAL PROXY INFORMATION

Our theoretical findings show that the global re-balance strategy works better than the local re-balance strategy in FL. However, it may violate the privacy protection requirements in FL. To overcome this drawback, we propose a class prior called the Global Proxy Information (GPI) based on

the model updates of the clients, inspired by an empirical observation: in the FC layer, the neuron weight of majority classes is updated more frequently than minority classes on the imbalance data. We further provide a theoretical guarantee that the label distribution cannot be observed through the numerical LPI value from the global perspective. Thus, GPI can be combined with existing re-balance strategies under privacy protection, which requires no additional information from clients.

**Definition 2** *(Local Proxy Information). We denote the gradient of $w^{ij}$ after one local optimization loop on $\mathcal{D}^k$ as $\nabla_{\mathcal{D}^k} \mathbf{W}^{ij}_{-1}$ and the dimension of the input of the FC layer as $H$. Then the local proxy information of class $i$ on $\mathcal{D}^k$ is defined as the gradient magnitude which is associated with $i$-th neuron of the FC layer:*

$$\Omega_i^k = \sum_{j=1}^{H} -\nabla_{\mathcal{D}^k} \mathbf{W}^{ij}_{-1}, \quad where \quad \nabla_{\mathcal{D}^k} \mathbf{W}^{ij}_{-1} = \frac{\partial F_k(\theta)}{\partial w^{ij}}. \tag{5}$$

Intuitively, local proxy information (LPI) accumulates the updates of the corresponding neurons of the last FC layer over the local dataset. Note that LPI is calculated without requirements for label distributions of the clients and the central server collects no additional information compared to the vanilla FL algorithm. We consider that the local training uses cross-entropy loss. As a motivation, we claim the theoretical evidence of the privacy protection of LPI in the following.

**Theorem 2** *For class $i$ in a local dataset $\mathcal{D}^k$, LPI implicitly connects with its label frequency (sample number in this class) and the probability that the sample is predicted to be the current class:*

$$\Omega_i^k \approx \frac{1}{n^k}(n_i^k - \mathcal{Z}_i) \sum_{(x,y) \in \mathcal{D}^k} \sum_{j=1}^{H} s_j, \quad where \quad \mathcal{Z}_i = \sum_{(x,y) \in \mathcal{D}^k} z_i, \tag{6}$$

where $s_j$ is $j$-th input of the FC layer, $y_i$ is $i$-th term of the one-hot ground truth, and $z_i$ is the probability of being predicted as class $i$. Theorem 2 indicates that the numerical value of LPI is implicitly relevant to the residual between the label frequency and probability from the local model, hence the local label distributions cannot be derived from the LPI values uploaded by the clients.

**Definition 3** *(Global Proxy Information). The global proxy information of class $i$ is defined as the weighted summation of the local proxy information:*

$$\Omega_i := \sum_{k=1}^{K} \frac{n^k}{n} \mathbb{I}(\Omega_i^k > 0)\Omega_i^k, \tag{7}$$

*where $\mathbb{I}(\cdot)$ is the indicator function and the value is 1 if $\cdot$ is true, 0 otherwise.*

$\mathbb{I}(\cdot)$ is necessary since negative LPI may result in unstable behavior of GPI. In fact, GPI is the gradient accumulation over the global dataset by collecting the LPI of the clients, which only involves the gradient information over local datasets. Importantly, we empirically find that GPI for re-balance strategies can alleviate the class imbalance issue while maintaining the recognition performance for majority classes, as shown in the experiment section 4.

During standard training in the practical FL setting, mini-batch SGD is commonly used for local training, where GPI can be approximately measured by accumulating gradients over all local datasets after several epochs of gradient descent steps. Besides, in each communication round, it is difficult to obtain an accurate GPI, since only a fraction of clients are selected for local updates. Thus, the global server calculates GPI based on the uploads of all clients only in the beginning communication round, then keeps a balanced training for subsequent rounds.

### 3.3 LABEL-DISTRIBUTION-AGNOSTIC ENSEMBLE LEARNING

Guided by our proxy information analysis, we propose a group-based ensemble learning framework to further handle the heterogeneous imbalance issue for FL. Our method separates clients into different groups based on their LPI. Models trained on individual groups are integrated together with a multi-expert structure, where each expert focuses on a group of clients with similar LPI. The overall

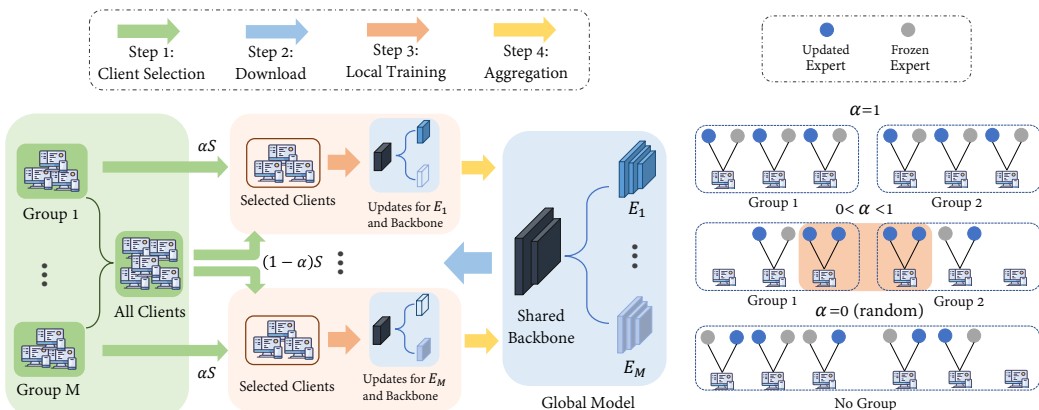

(a) Training pipeline of LDAE.      (b) Client selection with different $\alpha$

Figure 2: An overview of our group-based ensemble learning framework. (a) The training pipeline with four steps in a communication round: 1) For each expert, $\alpha S$ clients are randomly selected from the corresponding group and $(1 - \alpha)S$ clients are selected from the other clients. 2) The selected clients download the global model. 3) Each selected client updates the backbone and associated experts. 4) The server receives the updates and aggregates them into a new global model. (b) Cases with different $\alpha$ values for client selection. When $\alpha = 0$, clients in a group only train the corresponding expert. When $0 < \alpha \leq 1$, some clients may be chosen to update multiple experts.

framework and the model structure are shown in Figure 2(a). The global and local models have the same structure, *i.e.*, multiple experts with a shared backbone, and each expert has individual learnable blocks.

**Client Grouping.** Based on the section 3.2, the similarity score between LPI and GPI for each client can be calculated via the cosine distance, allowing us to monotonically rank the local clients. Then the ranked clients with close similarity scores are divided into the same group. Let hyper-parameter $M$ be the number of groups, then $M$ experts are allocated for these groups respectively, *i.e.*, group $P_i$ corresponds to expert $E_i$.

**Client Selection.** In one communication round, two parts of the clients are selected to update the same expert: (1) $\alpha S$ clients are randomly selected in each group ($P_i$), and (2) $(1 - \alpha)S$ clients are randomly selected from all clients other than clients in (1). $0 \leq \alpha \leq 1$ is a hyper-parameter to control the client selection. Empirically, the difference between experts can be controlled by the value of $\alpha$. As shown in Figure 2(b), if $\alpha = 1$, there is no interaction among the clients from different groups, where each group only updates the corresponding expert. If $\alpha = 0$, we adopt $M$ times boostrap sampling from all clients, where the clients selected at $i$-th time are used to update $E_i$ individually. As such, the experts are actually updated on the same global dataset and tend to be similar after sufficient rounds.

**Expert Ensembling.** After the client selection step, the selected clients download the current global model parameters and perform local training on their own datasets. In particular, for each client, the classification loss is calculated based on the average logits of all experts to interact the knowledge of different experts. Here, we adopt the balanced softmax loss (BSM) (Ren et al., 2020) for local training due to the effectiveness of logit adjusted loss for dealing with the class imbalance issue (Menon et al., 2020). We denote the class prior information, *e.g.*, global label distribution or GPI, as $\pi$, then the BSM is:

$$L_{\text{BSM}} = \frac{1}{n_k} \sum_{(x,y) \in \mathcal{D}_k} -y \log \sigma \left( \frac{1}{M} \sum_{i=1}^{M} \mathbf{v}_i(x, \theta) + \log \pi \right), \quad (8)$$

where $\sigma(\cdot)$ is the softmax function and $\mathbf{v}_i(\cdot)$ is the output logits of expert $E_i$. Notice that in the local training of one expert, only the parameters of the backbone and corresponding expert are updated,

while other experts are kept frozen. After receiving all the local updates, the server aggregates them into the global model with FedAvg McMahan et al. (2017).

Table 1: Top-1 accuracy on CIFAR-10-LT, CIFAR-100-LT, and ImageNet-LT. The GPI is combined with various class prior based re-balance algorithms to address the imbalance problem in FL.

| Methods | CIFAR-10-LT | | CIFAR-100-LT | | ImageNet-LT |
|---|---|---|---|---|---|
| | 100 | 50 | 100 | 50 | |
| FedAvg | 53.16 | 61.67 | 34.48 | 36.84 | 33.80 |
| FedProx | 61.43 | 71.78 | 34.16 | 38.77 | 32.98 |
| Ratio Loss | 53.31 | 62.26 | 33.06 | 34.94 | 33.15 |
| CLIMB | 60.28 | 72.29 | 34.66 | 40.22 | 35.29 |
| Focal Loss | 53.88 | 59.00 | 33.88 | 38.03 | 32.85 |
| CRT-IB | 53.80 | 63.52 | 32.48 | 37.61 | 31.77 |
| CRT-CB | 63.31 | 69.87 | 34.06 | 39.60 | 35.52 |
| LDAM + GPI | 66.73 | 72.58 | 35.36 | 39.17 | 34.88 |
| BSM + GPI | 67.44 | 74.36 | 37.19 | 41.91 | 37.64 |
| LADE + GPI | 64.89 | 71.79 | 38.61 | 41.08 | 38.52 |
| RIDE + GPI | 58.64 | 68.63 | 35.88 | 39.39 | 40.30 |
| PaCo + GPI | 68.84 | 75.35 | 37.76 | 43.24 | 39.04 |
| LDAE + GPI (ours) | **71.07** | **76.85** | **40.42** | **45.16** | **45.75** |

## 4 EXPERIMENT

**Datasets and Setup.** We conduct experiments on three long-tailed classification datasets: CIFAR-10-LT, CIFAR-100-LT (Cao et al., 2019) and ImageNet-LT (Liu et al., 2019). Following (Cao et al., 2019), we construct CIFAR-10-LT and CIFAR-100-LT by an exponential decay with the controllable IR (imbalance ratio). We show the experimental results with IR = 50 and IR = 100. We adopt Resnet-18 (He et al., 2016) for CIFAR-10-LT , Resnet-32 for CIFAR-100-LT and ResneXt for ImageNet-LT.

To simulate the data heterogeneity in FL, we divide the samples of each class in the global dataset into different clients according to the Dirichlet distribution (He et al., 2020; Yurochkin et al., 2019). Specifically, we first generate $\mathbf{p}_c \sim Dir_K(\alpha_{dir})$ for class $c$ and then allocating the $p_{c,k}$ proportion of the samples in class $c$ to client $k$. The degree of data heterogeneity is controlled by $\alpha_{dir}$, the parameter of the Dirichlet distribution. A small $\alpha_{dir}$ indicates high data heterogeneity. We set $\alpha_{dir}$ as 0.5 for CIFAR-10-LT, 0.1 for CIFAR-100-LT and 0.05 for ImageNet-LT. In each communication round, 20 clients are selected, and each client trains 2 local epochs. More implementation details are reported in Appendix A.

**Evaluation Protocol.** We evaluate the global model on the balanced dataset using top-1 accuracy. Following typical long-tail recognition tasks Liu et al. (2019), we categorize the classes into three groups: Many-shot class with $> 100$ samples, Medium-shot class with $\geq 20$ and $\leq 100$ samples and Few-shot class with $< 20$ samples.

### 4.1 RESULTS

**Comparison with Previous Methods.** To evaluate the effectiveness of our group-based multi-expert model, we conduct experiments comparing it with previous methods including the baseline algorithm FedAvg (McMahan et al., 2017), the method for dealing with data heterogeneity FedProx (Li et al., 2020a), methods for dealing with class imbalance in FL Ratio Loss(Wang et al., 2021) and CLIMB(Shen et al., 2021), and centralized long-tailed learning methods Focal Loss (Lin et al., 2017), CRT-IB, CRT-CB (Kang et al., 2019), LDAM (Cao et al., 2019), BSM (Ren et al., 2020), LADE (Hong et al., 2021), RIDE (Wang et al., 2020), and PaCo (Cui et al., 2021). The test accuracy on three typical long-tailed datasets is summarized in Table 1. Our approach significantly outperforms previous works on each dataset. Notice that the re-balancing methods requiring class prior information utilize our GPI. They have better results than other previous methods, especially on CIFAR-100-LT and ImageNet-LT, which have a large number of classes. The subsequent experiments show that the GPI makes a considerable contribution to their good performance.

Table 2: Comparison the performance of different re-balance strategies with local re-balance, global re-balance and GPI on CIFAR-10-LT, CIFAR-100-LT and ImageNet-LT.

| Methods | | CIFAR-10-LT | | CIFAR-100-LT | | ImageNet-LT |
| --- | --- | --- | --- | --- | --- | --- |
| | | 100 | 50 | 100 | 50 | |
| Local Re-balance | FedAvg | 53.16 | 61.67 | 34.48 | 36.84 | 33.80 |
| | LDAM | 63.55 | 69.50 | 32.54 | 37.01 | 30.70 |
| | BSM | 66.64 | 74.29 | 35.55 | 39.50 | 34.39 |
| | LADE | 66.05 | 74.54 | 33.28 | 39.89 | 36.32 |
| | RIDE | 59.95 | 69.05 | 34.94 | 39.09 | 40.00 |
| | PaCo | 65.11 | 71.35 | 35.92 | 40.03 | 38.26 |
| Global Re-balance | LDAM | 67.69 | 72.41 | 36.61 | 40.01 | 38.66 |
| | BSM | 65.81 | 74.15 | 38.56 | 42.45 | 39.63 |
| | LADE | 66.03 | 74.06 | 39.82 | 42.95 | 39.37 |
| | RIDE | 60.37 | 68.39 | 34.59 | 39.35 | 40.81 |
| | PaCo | 69.60 | 72.36 | 37.63 | 41.26 | 39.74 |
| | Ours | 69.76 | 75.12 | **41.77** | **45.32** | 45.57 |
| GPI | LDAM | 66.73 | 72.58 | 35.36 | 39.17 | 34.88 |
| | BSM | 67.44 | 74.36 | 37.19 | 41.91 | 37.64 |
| | LADE | 66.89 | 74.79 | 38.61 | 41.08 | 38.52 |
| | RIDE | 58.64 | 68.63 | 35.88 | 39.39 | 40.30 |
| | PaCo | 68.84 | 75.35 | 37.76 | 43.24 | 39.04 |
| | Ours | **71.07** | **76.85** | 40.42 | 45.16 | **45.75** |

**Effectiveness of GPI.** To further verify the effectiveness of GPI, we use local label distribution (local re-balance), global label distribution (global re-balance), and GPI as the class prior information for several typical centralized long-tailed methods. As shown in Table 2, GPI exhibits better performance than local re-balance and comparable results to global re-balance, indicating that our method can handle the imbalance issue for FL without data privacy leakage.

Table 3: Top-1 accuracy of re-balance strategies with local re-balance, global re-balance and GPI on Many/Medium/Few classes in CIFAR-100-LT with IR = 100.

| Methods | Local Re-balance | | | Global Re-balance | | | GPI | | |
| --- | --- | --- | --- | --- | --- | --- | --- | --- | --- |
| | Many | Medium | Few | Many | Medium | Few | Many | Medium | Few |
| FedAvg | 62.03 | 32.26 | 4.93 | - | - | - | - | - | - |
| LDAM | 56.23 | 29.91 | 7.97 | 52.03 | 37.63 | 17.43 | 56.14 | 33.80 | 12.93 |
| BSM | 57.63 | 36.34 | 8.87 | 51.54 | 41.49 | 20.00 | 52.80 | 39.26 | 19.07 |
| LADE | 54.00 | 33.74 | 8.57 | 52.57 | 43.34 | 20.83 | 55.34 | 40.46 | 12.20 |
| RIDE | 64.97 | 31.11 | 4.37 | **63.29** | 32.34 | 3.73 | **63.66** | 34.91 | 4.60 |
| PaCo | 35.60 | 46.34 | 24.13 | 42.06 | 41.23 | **28.27** | 45.94 | 42.23 | 22.67 |
| Ours | - | - | - | 50.29 | **44.71** | 28.07 | 49.97 | **44.14** | **24.93** |

Furthermore, we report the accuracy of many/medium/few-shot classes on CIFAR-100-LT with IR = 100 in Table 3. Compared with local re-balance using local label distribution, previous cost-sensitive re-balance methods (*i.e.*, LDAM, Balanced SM and LADE) with GPI can achieve significant improvements for medium-shot and few-shot classes. Similarly, benefiting from GPI, RIDE and PaCo also further increases the recognition accuracy for medium-shot and many-shot classes. The slight performance improvement of few-shot classes may be attributed to the fact that RIDE and PaCo focus on representation quality over all classes rather than merely minority classes. On the other hand, compared with global re-balance, GPI shows competitive results for all classes, although it exhibits relatively lower accuracy for the medium-shot and few-shot classes. Notice that GPI performs better on many-shot classes sometimes due to the subtraction of the probability term, as we mentioned in section 3.2. Compared with the previous SOTA method RIDE, our method obtains higher accuracy since heterogeneous imbalance is alleviated via client grouping.

**GPI Visualization.** As shown in Figure 3, we visualize the class-wise GPI curves on CIFAR-100-LT with IR = 10 and IR = 100. Compared with the groundtruth label distribution on the same IR,

GPI curves exhibit a similar tendency, which can be used for class prior based re-balance strategies. Besides, GPI can maintain the performance of majority classes by using smaller GPI values for re-balancing. It should be noticed that GPI is merely obtained from the model updates implicitly combining label frequency and prediction probability as mentioned in Theorem 2.

**Ablation Studies on the Ensemble Framework.** Table 4 shows the component analysis of the proposed ensemble framework. We compare three different grouping methods: grouping clients randomly (Random Grouping), grouping clients by the cosine distances between local and global label distributions (LD Grouping), and grouping clients by the cosine distances between LPI and GPI (PI Grouping). The results show that Random Grouping contributes little to the performance, while LD Grouping and PI Grouping bring about 3% improvement in accuracy. Our grouping method can further improve the final performance of other re-balance strategies besides BSM (Ren et al., 2020), such as LDAM loss (Cao et al., 2019). Moreover, we evaluate the influence of different values of $\alpha$, the hyper-parameter to control the client selection scope of each expert. As shown in Figure 4, the performance can be enhanced via an appropriate value of $\alpha$.

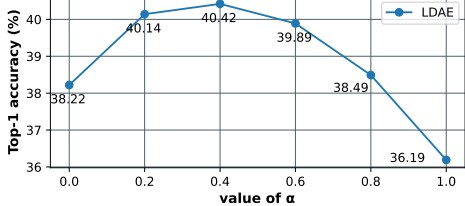

Figure 3: Comparison between the global label distribution and GPI on CIFAR-100-LT with IR = 10 and IR = 100.

| Methods | Accuracy |
|---|---|
| LDAM | 35.36 |
| BSM | 37.19 |
| PI Grouping + LDAM | 38.25 |
| Random Grouping + BSM | 37.38 |
| LD Grouping + BSM | 40.51 |
| PI Grouping + BSM (Ours) | 40.42 |

Table 4: Ablation studies for the grouping method and training loss of the ensemble framework on CIFAR-100-LT with IR = 100.

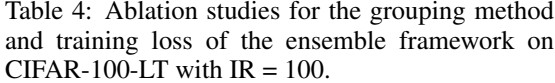

Figure 4: Ablation studies on different $\alpha$ for client selection in the ensemble framework on CIFAR-100-LT with IR = 100.

**Data Heterogeneity.** Existing re-balance strategies under the FL setting generally aim to train a balanced local model with compromising performance due to the heterogeneous property of FL. As shown in Table 1, Table 2 and Table 3, GPI can significantly improve the performance by alleviating the data heterogeneity issue. Moreover, our multi-expert framework could ensemble knowledge learned from different local data sources to further address the heterogeneous imbalance.

**Privacy Protection.** To train a balanced global model, class prior based re-balance methods mainly rely on global label distribution without privacy protection, while the proposed GPI is merely related to the model updates without extra label information. Thus, our method can maintain the data privacy of clients and simultaneously achieve better performance on all the benchmarks.

## 5 CONCLUSIONS

In this work, we focus on the long-tailed distribution problem in FL. Existing re-balance methods for FL suffer from the data heterogeneity issue and assume label distribution is available without considering privacy requirements. To this end, we propose a label-distribution-agnostic ensemble learning (LDAE) framework to re-balance the global objective with preserving privacy. In particular, we design the proxy information to guide the balanced training of multiple experts, which only involves model updates for preserving privacy. Extensive experimental results indicate the effectiveness of our method compared with previous SOTA methods.

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

# A   IMPLEMENTATION DETAILS

The number of communication rounds is 2000 for most methods on CIFAR-10/100-LT and 1000 on ImageNet-LT. On the CIFAR-100-LT, the learning rate is initialized as 0.5 and decayed by 0.05 at round 1600 for all methods except PaCo (Cui et al., 2021), where the initial learning rate is 0.3. On the CIFAR-10-LT (Cao et al., 2019) with imbalance ratio 50 and 100, the initial learning rate is 0.1 for FedAvg (McMahan et al., 2017), Ratio Loss (Wang et al., 2021), CLIMB (Shen et al., 2021), Focal Loss (Lin et al., 2017), CRT (Kang et al., 2019), LDAM (Cao et al., 2019), BSM (Ren et al., 2020), LADE (Hong et al., 2021), and LDAE. It is 0.2 for RIDE (Wang et al., 2020) and 0.3 for PaCo. The learning rate is decayed by 0.1 for all methods. With RIDE, PaCo, and LDAE, the model is trained for 2500 communication rounds on CIFAR-10-LT due to their complex model structures. For training on ImageNet-LT (Liu et al., 2019), we set the training round as 1000 and learning rate as 0.1, which is decayed at 800th round by 0.1. For CRT, we retrain the classifier on the last 200 rounds. For LADE, we set the weight of LADER as 0.01 for CIFAR-10-LT and 0.1 for CIFAR-100-LT and ImageNet-LT. We set $\alpha = 0.5$ in LDAE on all datasets.

# B   PROOFS

## B.1   PROOF TO THEOREM 1

We denote the global objective of FL model with global re-balance and local re-balance as $G_g(\theta)$ and $G_l(\theta)$, respectively, then we have:

$$
\begin{aligned}
G_g(\theta) &= \sum_{k \in \{0,1\}} \frac{n_k}{n_0 + n_1} \sum_{c \in \{+,-\}} \frac{n_0 + n_1}{(n_0^c + n_1^c) n_k} \sum_{(\boldsymbol{x}, y^c) \in \mathcal{D}^k} f(\boldsymbol{x}, y^c, \theta) \\
&= \sum_{c \in \{+,-\}} \frac{1}{n_0^c + n_1^c} \sum_{(\boldsymbol{x}, y^c) \in \mathcal{D}} f(\boldsymbol{x}, y^c, \theta) ,
\end{aligned}
\tag{9}
$$

and

$$
\begin{aligned}
G_l(\theta) &= \sum_{k \in \{0,1\}} \frac{n_k}{n_0 + n_1} \sum_{c \in \{+,-\}} \frac{1}{n_k^c} \sum_{(\boldsymbol{x}, y^c) \in \mathcal{D}^k} f(\boldsymbol{x}, y^c, \theta) \\
&= \sum_{c \in \{+,-\}} \frac{1}{n_0 + n_1} \left( \frac{n_0}{n_0^c} + \frac{n_1}{n_1^c} \right) \sum_{(\boldsymbol{x}, y^c) \in \mathcal{D}} f(\boldsymbol{x}, y^c, \theta) ,
\end{aligned}
\tag{10}
$$

where $c$ is the class index. Then, the objective gap is:

$$
\begin{aligned}
\Delta &= G_l(\theta) - G_g(\theta) \\
&= \sum_{c \in \{+,-\}} \frac{n_1 (n_0^c)^2 + n_0 (n_1^c)^2}{n_0^c n_1^c (n_0 + n_1)(n_0^c + n_1^c)} \sum_{(\boldsymbol{x}, y^c) \in \mathcal{D}} f(\boldsymbol{x}, y^c, \theta).
\end{aligned}
\tag{11}
$$

For sample number $n_k^c > 0$, we have $\Delta > 0$, indicating that the objective function value of local re-balance is always larger than global re-balance on the same dataset. Thus, let $\mathcal{E}$ be the biased estimation of global label distribution, there exists a group of re-balance weights $e$ derived from $\mathcal{E}$, where $e$ is close to the global re-balance strategy. Then re-balance weights $e$ based global objective $G_e$ satisfies:

$$
G_g(\theta) \le G_e(\theta) < G_l(\theta).
\tag{12}
$$

## B.2   PROOF TO THEOREM 2

Follow the definition 2, we have:

$$
\Omega_i^k = -\frac{\partial F_k(\theta)}{\partial w^{ij}}.
\tag{13}
$$

Then assuming cross-entropy loss are used, we can derive the following equation:

$$
\Omega_i^k = \frac{1}{n^k} \sum_{(x,y) \in D^k} (y_i - z_i) \sum_{j=1}^{H} s_j ,
\tag{14}
$$

where $s^j$ is $j$-th input of the FC layer, $y^i$ is $i$-th term of the one-hot ground truth, and $z^i$ is the probability of being predicted as class $i$. Since $\sum_{j=1}^{H} s_j$ is usually independent of the class, we can write the above equation as:

$$\Omega_i^k \approx \frac{1}{n^k}(n_i^k - \mathcal{Z}_i) \sum_{(x,y)\in\mathcal{D}^k} \sum_{j=1}^{H} s_j, \quad \text{where} \quad \mathcal{Z}_i = \sum_{(x,y)\in\mathcal{D}^k} z_i. \tag{15}$$

## C  COMPARISON OF FEDERATED LEARNING METHODS

Table 5: Top-1 accuracy of various FL algorithms for dealing with data heterogeneity on CIFAR-100-LT.

| IR | FedAvg | | FedProx | | SCAFFOLD | |
|---|---|---|---|---|---|---|
| | w/o BSM | w/ BSM | w/o BSM | w/ BSM | w/o BSM | w/ BSM |
| 50 | 36.84 | 41.91 | 38.77 | 42.33 | 40.44 | 42.27 |
| 100 | 34.48 | 37.19 | 34.16 | 37.78 | 34.70 | 38.55 |
| IR | FedAlign | | Ditto | | FedRep | |
| | w/o BSM | w/ BSM | w/o BSM | w/ BSM | w/o BSM | w/ BSM |
| 50 | 39.80 | 43.66 | 37.03 | 39.29 | 37.69 | 39.17 |
| 100 | 35.36 | 39.21 | 33.45 | 35.18 | 32.23 | 35.04 |

We further experimented with FedProx (Li et al., 2020a), SCAFFOLD (Karimireddy et al., 2020), FedAlign (Mendieta et al., 2022), Ditto (Li et al., 2021b) and FedRep (Collins et al., 2021) as baselines on the CIFAR100-LT. They are recent FL algorithms for dealing with data heterogeneities. When combined with BSM loss (Ren et al., 2020), for non-personalized FL methods (i.e., FedProx (Li et al., 2020a), SCAFFOLD (Karimireddy et al., 2020), FedAlign (Mendieta et al., 2022)), we use GPI as the class prior, and for personalized FL methods (i.e., Ditto (Li et al., 2021b), FedRep (Collins et al., 2021)), local re-balance is applied.

As shown in Table 5, the non-personalized FL methods are effective in FL with relatively mild class imbalance (imbalance ratio = 50), while are less effective when class imbalance is severe (imbalance ratio = 100). More importantly, when combined with our re-balance strategy, these methods can obtain a significant performance improvement for various imbalance ratios.

However, we test the personalized FL methods Ditto [4] and FedRep [5] on the federated long-tailed problem, and as the table above shows, they do not perform well on the federated long-tailed problem, even with more training rounds. We believe the main reason is the distribution shift. Under the general setting of personalized FL without the long-tailed problem, the distribution of test data and training data are identical. However, considering the long-tailed problem, the training data is imbalanced and the test data is balanced for each client, resulting in a distribution shift. The personalized model focuses more on fitting the local training data distribution and therefore generalizes poorly to different data distributions. Compared with the personalized FL methods, non-personalized FL methods with a well-trained global model have a stronger generalization ability, so they can achieve better performance on the federated long-tailed problem.

Table 6: Top-1 accuracy on CIFAR-100-LT and the model size of LDAE with various expert numbers.

| Expert Number M | CIFAR-100-LT IR = 100 | CIFAR-100-LT IR = 50 | Parameters(Million) |
|---|---|---|---|
| 1 | 37.19 | 41.91 | 0.46 |
| 2 | 38.71 | 44.92 | 0.52 |
| 3 | 40.42 | 45.16 | 0.77 |
| 4 | 41.68 | 46.88 | 1.02 |
| 5 | 41.85 | 46.59 | 1.27 |

# D    ABLATION STUDY FOR EXPERT NUMBER OF LDAE

We show the effect of expert number $M$ of our multi-expert method LDAE. A larger $M$ can lead to more parameters. The results indicate that as $M$ increases, the recognition accuracy improves. Considering the trade-off between performance and communication cost, 3 and 4 are both good values for $M$. When $M$ is larger than 4, the improvement in accuracy is slight.

# E    VISUALIZATION OF DATA DISTRIBUTION

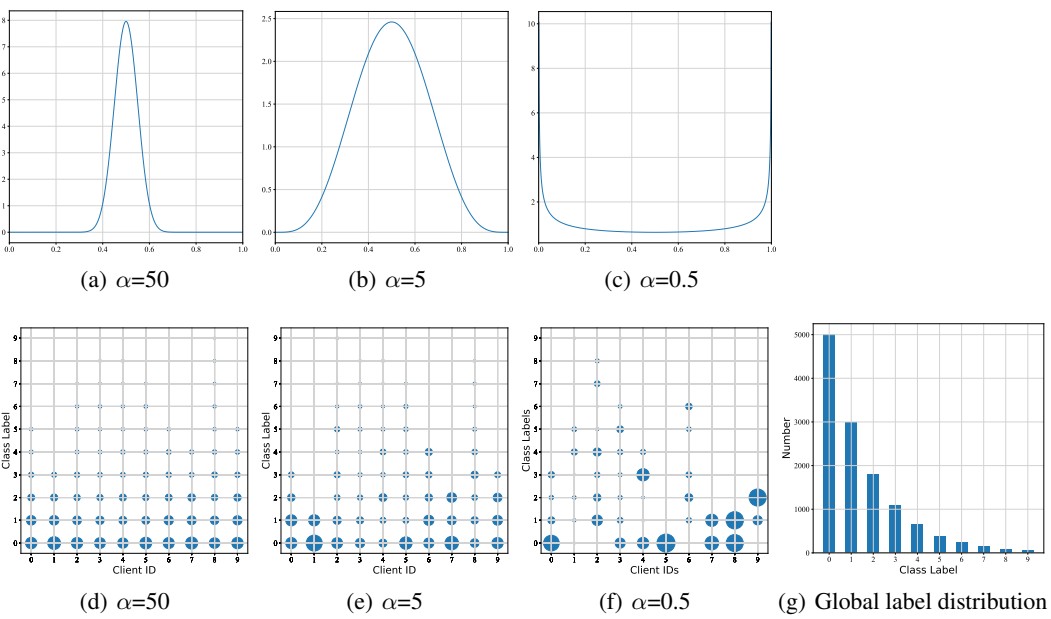

(a) $\alpha$=50    (b) $\alpha$=5    (c) $\alpha$=0.5

(d) $\alpha$=50    (e) $\alpha$=5    (f) $\alpha$=0.5    (g) Global label distribution

Figure 5: The data distributions of local datasets with different settings.

To simulate the data heterogeneity, for each class in the global dataset, we partition the samples into different clients according to the Dirichlet distribution. In this section, we partition the CIFAR10-LT into 10 clients with different values of $\alpha_{dir}$, i.e., the hyper-parameter of the Dirichlet distribution. In Figure 5, (d, e, f) show the data distributions of local clients with different values of $\alpha_{dir}$, and (a, b, c) show the corresponding probability density of the two-dimensional Dirichlet distribution. The point size indicates the sample number. Small $\alpha_{dir}$ results in higher heterogeneity. The global label distribution of the CIFAR10-LT with IR = 100 is shown in (g).

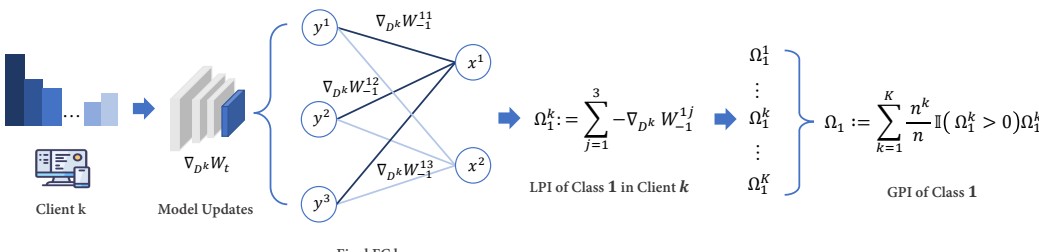

Figure 6: An example of LPI and GPI calculation process.

# F    EXAMPLE OF LPI AND GPI CALCULATION PROCESS

Figure 6 intuitively illustrates how LPI and GPI are calculated. In the first round, the server receives the model updates uploaded by each client. The LPI of class 1 for client $k$ is calculated by adding the updated values of the weights connected to neuron $x^1$ according to the reference local proxy. The GPI of class 1 is then the weighted sum of all clients' LPI based on the Eq.3. The GPI of other classes is calculated in the same way.

# G    VISUALIZATIONS OF Z VALUE, LPI, GPI, AND LABEL DISTRIBUTION

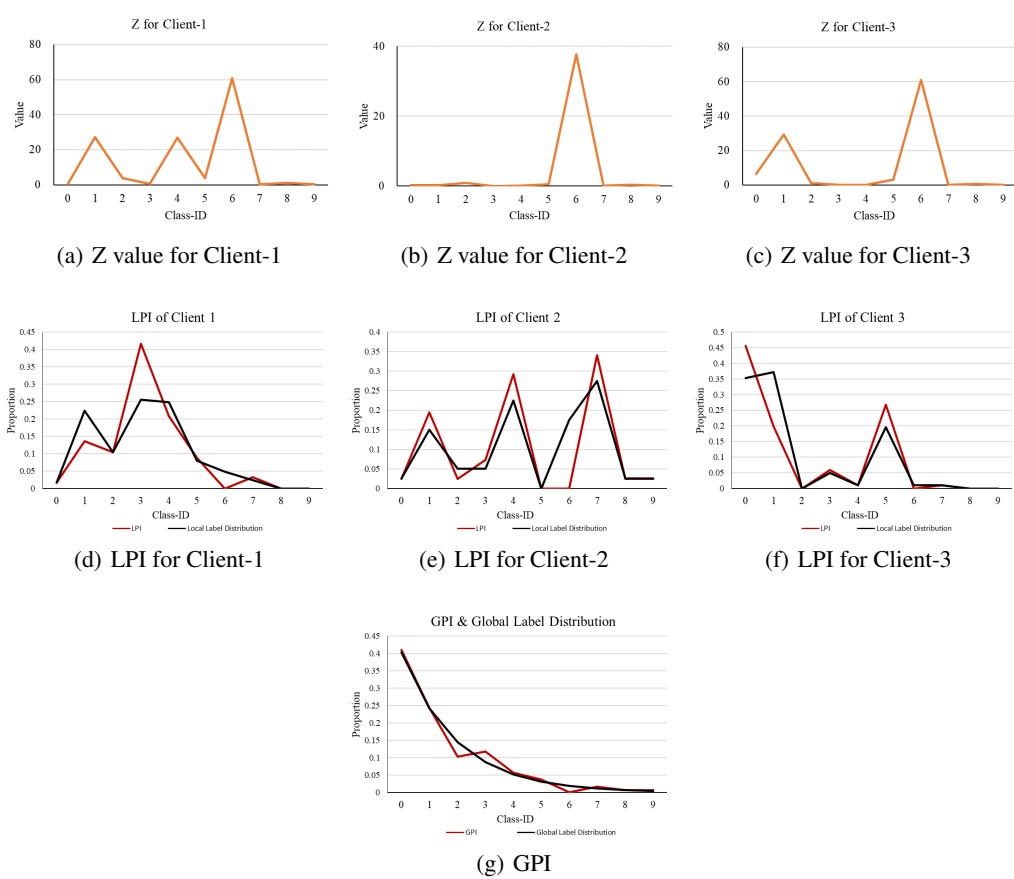

(a) Z value for Client-1       (b) Z value for Client-2       (c) Z value for Client-3

(d) LPI for Client-1       (e) LPI for Client-2       (f) LPI for Client-3

(g) GPI

Figure 7: Visualizations of $Z$ (Eq.6), LPI with local label distribution, GPI with global label distribution on CIFAR-10-LT with $\alpha = 0.5$. Sub-figure (a) and (d) are for client 1, (b) and (e) are for client 2, (c) and (f) are for client 3.

To verify Theorem 2, we conduct an empirical study on CIFAR-10-LT ($\alpha = 0.5$ and IR=100), which include $Z$ (Eq.6) value curve, LPI with local label distribution, and GPI with global label distribution. These values are obtained in the first communication round. The experimental setup is described in Appendix A. We randomly select 3 clients from 100 clients for visualizations.

**Z Value (Eq.6).** As shown in Sub-figure (a), (b), and (c) of Figure 7, we visualize the class-wise $Z_i$ values for the models of different clients at the initial communication round. it is observed that the model in the first communication round may make a biased prediction instead of uniform predictions. Therefore, the $Z_i$ obtained at this stage may also be biased.

**LPI with Local Label Distribution.** As shown in Sub-figure (d), (e), and (f) of Figure 7, we visualize the class-wise LPI estimated at the initial communication round and the groundtruth local label distribution for different clients. The LPI with a negative value is set to 0 since it will not be used in the calculation of GPI. As these Sub-figures show, even under the biased $Z_i$ values, the LPI curve for each client can exhibit a similar tendency to the corresponding local label distribution.

**GPI with Global Label Distribution.** As shown in Sub-figure (g) of Figure 7, we visualize the class-wise GPI estimated at the initial communication round. GPI curves have a similar tendency when compared with the groundtruth global label distribution (black curve). Thus, although $Z_i$ may be biased at the beginning communication round, the estimated GPI can still reflect the trend of global label distribution. Therefore, GPI can be used as the class prior for re-balance strategies under the biased prediction at the beginning communication round.

