# OpenReview forum: "Label-distribution-agnostic Ensemble Learning on Federated Long-tailed Data"
_ICLR.cc/2023/Conference — Submitted to ICLR 2023_

### Official Review · Reviewer_prdG · 2022-10-17

**Confidence:** 5
**Correctness:** 3
**Technical Novelty And Significance:** 3
**Empirical Novelty And Significance:** 3
**Recommendation:** 6

**Clarity, Quality, Novelty And Reproducibility:**

Clarity: the writing of the method is hard to follow.
Quality: the quality is overall good, but there are a few issues that should be resolved.
Novelty: novelty is fair.
Reproducibility: the code is expected to release by the authors.

**Strength And Weaknesses:**

Strengths:
1. The studied task is practical and highly challenging.
2. Exploring privacy-preserving proxy to guide model training is new to me.
3. The empirical results seem promising.

Weaknesses:
1. Introduction is good-writing, but the method section is hard to follow.
2. Please carefully check the proof of Theorem 2. It seems not correct from Eq.12 to Eq.13.
3. please clarify why local label distributions cannot be derived from the LIP values if it is relevant to the residual.
4. It is unclear to me why the estimated GPI based on the beginning communication round can keep balanced prior for guiding model training.
5. For global proxy information, how can you know n^k, i.e., the class-level sample number in each client?
6. Please ablate the expert number M.

**Summary Of The Paper:**

This paper studies Federated Long-tailed Data, which is a highly challenging task. To address this task, this paper proposes to use a privacy-preserving proxy to guide model training and develops a label-distribution-agnostic ensemble learning framework. Both theoretical analysis and empirical verification are provided.

**Summary Of The Review:**

I like this paper overall, but there are a few issues that need to be addressed before acceptance.

**********Post rebuttal***************
Thanks a lot for the response. Most of my concerns have been clarified. Even so, I still have some concerns about the biased prediction at the early stage. However, it is just a minor concern, so I keep my original score of 6 but increase my confidence to 5.

---

> ### Author Response · Authors · 2022-11-18
> **Response to Reviewer prdG**
>
> Dear Reviewer prdG,
>
> Thank you for your valuable time and thoughtful comments. The code is available at https://anonymous.4open.science/r/anonymous-001/. Below, we address all the points you raised.
>
> ---
>
> **Q1: Please carefully check the proof of Theorem 2. It seems not correct from Eq.12 to Eq.13.**
>
> **R1**: Thank you sincerely for the correction. We have double-checked the proof process and there was indeed a typo in Eq. (12). We have corrected it in our revised manuscript (please see Appendix B.2).
>
> **Q2: Please clarify why local label distributions cannot be derived from the LPI values if it is relevant to the residual.**
>
> **R2**: Thanks for your comment. Since the samples are always on the local clients, the exact values of $Z_i$ and $s_i$ associated with the samples in Eq. (6) are always unknown to the server. Thus the server cannot obtain the exact local label distribution directly from the LPI without any additional information such as a small balanced dataset from the server.
>
> **Q3: It is unclear to me why the estimated GPI based on the beginning communication round can keep balanced prior for guiding model training.**
>
> **R3**: Thanks for the comment. We would like to explain it in the following two aspects:
>
> - At the beginning communication round, the model does not yet fit the training data well. In this case, the model usually makes a uniform prediction for different classes, resulting in an approximately uniform distribution of $Z_i$ in Eq. (6) for different classes. Based on uniformly distributed $Z_i$, LPI can accurately reflect the relative trend of the local label distribution according to Eq. (6), then GPI can be a good estimation for the trend of the global label distribution. Therefore, the GPI estimated at the beginning communication can be a better class prior for the re-balance methods than the updating GPI along with training.
> - On the other hand, GPI calculation requires gradient information over all clients. Updating GPI during the training will result in more communication costs.
>
> **Q4: For global proxy information, how can you know n^k, i.e., the class-level sample number in each client?**
>
> **R4**: Sorry for the unclear description. Indeed, $n^k$ is not the class-level sample number of each client. It denotes the total number of samples for the client $k$, which is required by the FedAvg algorithm to do a weighted average of the model updates. The class-level sample number is unknown to the server. In **Notation** on page 3, we add the description of $n^k$ to "Each client $k$ has a local training dataset $\mathcal{D}^{k}$ with a total of $n^k$ samples.".
>
> **Q5: Please ablate the expert number M.**
>
> **R5**: Thanks for your good suggestion. We add the ablation study for expert number M. The results indicate that the recognition accuracy is improved along with increasing M. However, a larger M may lead to more parameters. Considering the trade-off between performance and communication cost, 3 and 4 are both reasonable values of M. When M is larger than 4, the improvement in accuracy is not significant.
>
> | Expert Number M | Imbalance ratio 100 | Imbalance ratio 50 | Parameters (Million) |
> | --------------- | ------------------- | ------------------ | -------------------- |
> | 1               | 37.19               | 41.91              | 0.46                 |
> | 2               | 38.71               | 44.92              | 0.52                 |
> | 3               | 40.42               | 45.16              | 0.77                 |
> | 4               | 41.68               | 46.88              | 1.02                 |
> | 5               | 41.85               | 46.59              | 1.27                 |

---

> > ### Comment · Reviewer_prdG · 2022-11-18
> > **Further discussion**
> >
> > Thanks for the response. I would like to add one follow-up question. First, regarding Q3, I see your point. At the beginning communication round, the model does not yet fit the training data well and thus has an approximately uniform prediction. However, it may also be a biased prediction and cannot represent the true class distribution. How does the proposed method deal with this issue in real applications?

---

> > > ### Author Response · Authors · 2022-11-18
> > > **Response to Reviewer prdG**
> > >
> > > Many thanks for your review and the further comment. To answer your question, we experimentally verify whether the model at the initial communication round would make a biased prediction, i.e. the value of $Z_i$ in Eq. (6), and how it affects the value of LPI and GPI. The main results are shown in Appendix G of the revised manuscript.
> > >
> > > - It is observed that the model may make a biased prediction at the initial communication round. We discuss this as following two points.
> > >
> > >   (1) Influenced by imbalance training, the model tends to predict a larger Z for the classes with larger sample numbers in a client. In this case, we find that the proportion of LPI for these classes will be smaller than the real proportion of this class. However, since these classes usually represent a large proportion, the effect of this bias on the trend of the LPI is not particularly severe. Thus, the LPI of this class can still closely reflect the real proportion of this class. For example, classes 1 and 4 of client 1 have a larger value of Z as shown in sub-figure (a), and the estimations of LPI for them are still close to the local label distribution as shown in sub-figure (d).
> > >
> > >   (2) Influenced by random initialization, the model tends to predict a larger $Z_i$ for a particular class $i$. For example, class 6 of clients 1, 2, and 3 in sub-figure (a), (b), and (c), respectively. If this class has a large number of samples, the case is the same as (1). If this class only has a few samples, the LPI and GPI of this class are influenced, while other classes can still obtain a good estimation of LPI and GPI. Thus, the total trend GPI is not affected much, especially when the class number is large.
> > >
> > > - We also report the cosine similarity between LPI and the local label distribution for Client 1, Client 2, and Client 3, as well as between GPI and the global label distribution. The results show that GPI can reflect a very close trend to the global label distribution, although the connection between LPI and local label distribution is slightly weaker than it is between GPI and global label distribution.
> > >
> > > | Proxy Information-Label Distribution | Cosine Similarity |
> > > | :----------------------------------: | :---------------: |
> > > |               Client 1               |      0.9247       |
> > > |               Client 2               |      0.9106       |
> > > |               Client 3               |      0.9265       |
> > > |                Global                |      0.9937       |

---

> > > > ### Comment · Reviewer_prdG · 2022-11-19
> > > > **Discussion**
> > > >
> > > > Thanks for the new response. It is clearer. Moreover, how do both LPI and GPI change during model training? In addition, is the observation consistent with ImageNet-LT?

---

> > > > > ### Author Response · Authors · 2022-11-22
> > > > > **Response to Reviewer prdG**
> > > > >
> > > > > Thanks for your inspiring comments. We would like to answer them from the following two aspects.
> > > > >
> > > > > - **How does the GPI/LPI change during the training?**
> > > > >
> > > > >   The training process can be divided into two stages and we show the analysis based on the Theorem 2 (i.e., Eq. (6) in the revised manuscript):
> > > > >
> > > > >   1. In the early training stage, the absolute values of all classes' GPI/LPI are large. Specifically, as analyzed in the previous response, there is a biased prediction towards one particular class in each client, corresponding to a large $Z_i$ for that class, while small $Z_i$ values for other classes. Thus, the LPI of the biased class is negative, while most of the other classes' GPI/LPI are positive.
> > > > >   2. In the remaining training stage, GPI/LPI's absolute value of all classes becomes smaller, as the $Z_i$ becomes relatively uniform. The GPI/LPI of tail classes (classes with small sample numbers) is small in this stage due to the small $n_i$, while the GPI/LPI of head classes (classes with large sample numbers) is relatively large. As the training continues, the model learns head classes well and predicts large $Z_i$ for them. Thus, the gradients w.r.t. these classes (i.e., GPI/LPI) are small.
> > > > >
> > > > >   **In summary**, the GPI/LPI of each class changes according to $Z_i$, which is firstly influenced by the biased prediction, and then influenced by the updated model, depending on the global class distribution and the learning algorithm, etc. More details and analysis about the changing of GPI/LPI will be added in the final revision.
> > > > >
> > > > > - **How does cosine similarity between GPI/LPI and global/local label distribution change during training?**
> > > > >
> > > > >   We show the cosine similarity between GPI/LPI and global/local label distribution at some rounds in the following table. LPI Avg denotes the average of the similarity between LPI and the local label distribution for all clients.
> > > > >
> > > > >   - Consider FL without re-balance (FL).
> > > > >
> > > > >     - For CIFAR10-LT, the similarity decreases after the first round. The GPI is close to the label distribution after the first round, as shown in the previous response. The GPI of head classes tends to be small as the training progresses, so the difference with the label distribution grows larger and larger.
> > > > >
> > > > >     - For CIFAR100-LT and ImageNet-LT, the cosine similarity first increases and then decreases. Since their head classes are more difficult to learn than CIFAR10-LT, the GPI of head classes is greater at the early stage. Therefore, as it decreases in stage 2, the overall GPI will approach the real label distribution before moving away. We only have results for 300 rounds on ImageNet-LT now because the calculation of the GPI needs the LPI of all clients and is time-consuming.
> > > > >
> > > > >   - Consider using BSM loss with our GPI for re-balance (BSM). The change of similarity has almost the same trend as training without re-balance. However, the value of similarity is larger with re-balance since $Zi$ is more uniform.
> > > > >
> > > > >   **Overall**, the similarity is high enough at the beginning for different datasets, although it may rise during training on CIFAR100-LT and ImageNet-LT. Considering the communication cost, it is reasonable to compute GPI in the first round.
> > > > >
> > > > > | Cosine Similarity |             |              |               |                |
> > > > > | :---------------: | :---------: | :----------: | :-----------: | :------------: |
> > > > > |  **CIFAR10-LT**   | **Round 1** | **Round 10** | **Round 100** | **Round 1000** |
> > > > > |     GPI (FL)      |    99.37    |    95.80     |     94.41     |     90.21      |
> > > > > |   LPI Avg (FL)    |    92.84    |    86.42     |     88.38     |     73.62      |
> > > > > |     GPI (BSM)     |    99.37    |    96.63     |     98.54     |     97.40      |
> > > > > |   LPI Avg (BSM)   |    92.84    |    91.67     |     96.53     |     95.81      |
> > > > > |  **CIFAR100-LT**  | **Round 1** | **Round 10** | **Round 100** | **Round 1000** |
> > > > > |     GPI (FL)      |    98.39    |    98.85     |     98.87     |     96.81      |
> > > > > |   LPI Avg (FL)    |    96.36    |    97.52     |     98.02     |     96.17      |
> > > > > |     GPI (BSM)     |    98.39    |    98.93     |     99.63     |     98.17      |
> > > > > |   LPI Avg (BSM)   |    96.36    |    97.84     |     98.93     |     96.55      |
> > > > > |  **ImageNet-LT**  | **Round 1** | **Round 10** | **Round 100** | **Round 300**  |
> > > > > |     GPI (FL)      |    91.39    |    94.80     |     94.77     |     93.49      |
> > > > > |   LPI Avg (FL)    |    85.83    |    88.21     |     88.58     |     89.07      |
> > > > > |     GPI (BSM)     |    91.39    |    95.31     |     97.20     |     96.50      |
> > > > > |   LPI Avg (BSM)   |    85.83    |    92.10     |     94.03     |     93.88      |

---

> > > > > > ### Comment · Reviewer_prdG · 2022-11-25
> > > > > > **Response to Authors**
> > > > > >
> > > > > > Thanks a lot for the response. My concerns have been clarified better. Even so, I still have some concerns about the biased prediction at the early stage. However, it is just a minor concern, so I keep my original score of 6 but increase my confidence to 5. Good luck.

---

> > > > > > > ### Author Response · Authors · 2022-11-25
> > > > > > > **Response to Reviewer prdG**
> > > > > > >
> > > > > > > Dear Reviewer prdG,
> > > > > > >
> > > > > > > We are highly encouraged by your positive comments. The change of biased prediction is really a phenomenon worth exploring in depth. We will add more details and analysis about it in the final revision. Many thanks for your constructive comments.
> > > > > > >
> > > > > > > Best Regards,
> > > > > > > The Authors

---

### Official Review · Reviewer_5sfY · 2022-10-24

**Confidence:** 4
**Correctness:** 2
**Technical Novelty And Significance:** 2
**Empirical Novelty And Significance:** 2
**Recommendation:** 5

**Clarity, Quality, Novelty And Reproducibility:**

Novelty & Quality:

The proposed algorithm is novel to me. It is quite interesting (although there is no theoretical analysis) how the clients can be clustered based on their (unknown) label distribution using the LPI & GPI.

However, I have the following reservations regarding the contribution claims (some of which were mentioned above in Weaknesses):

- Section 3 shows how a certain global rebalancing technique will induce the same result regardless of whether the setting is federated or centralized; it also shows there is a difference between the ensuing loss if the same rebalancing technique is instead applied at local sites. I agree with all those results but I do not follow how these explain for the improved performance of global rebalancing over local rebalancing: being different does not explain why one is better than the other. In my opinion, the delta term in Theorem 1 needs more intuitive interpretation. Otherwise, the current one reads local re-balance is worse than global re-balance because they are different, which (in my opinion) is not a valid explanation.

- Regarding the privacy claim: if clients can be clustered in terms of their label distribution similarity using the LPI then it is actually not safe to say the LPI cannot be used to infer the local label distribution -- privacy might not be preserved if the server has access to a small (balanced) validation set that could be used to probe the performance of each client cluster, which will reveal what are their label distributions; in addition, the definition of LPI also suggests that once the training converged, the Z and s terms in Eq. (5) will become more or less fixed and so, the server might be able to solve a linear system to assess the local label distribution -- it might not be exact depending on whether the system is under-constrained but it will still reveal extra information about those label distributions so the claim of privacy preservation needs to be revisited.

- For the same problem, personalized federated learning could be applied as a viable solution technique. The authors should discuss this; and provide empirical comparison to advocate for the practical significance of the proposed solution.

Clarity:

- The writing is pretty good and has conveyed the key idea effectively. However, some algorithmic details still remain vague. For instance, how do the rebalance techniques used in baselines get incorporated into the FL pipeline & which one is the strategy analyzed in Section 3?

- It is also strange local clients with negative LPI were excluded from the aggregation that produces GPI -- can the authors elaborate more?

- How are the hyper-parameters such as M and alpha selected?

Reproducibility:

The algorithm is well-described so I believe its reproducibility. Is the code released somewhere? Reproducibility is strengthened if the code is also released.

**Strength And Weaknesses:**

Strengths:

+ The problem is well-motivated; the presentation of the proposed algorithm is well-structured.
+ The algorithm is tested on multiple large-scale benchmark datasets for image classification (CIFAR-100, ImageNet)
+ The proposed algorithm appears novel & reportedly performs better than the baselines

Weaknesses:

- Lacking coverage of other types of FL algorithms that deal with data heterogeneities, such as personalized federated learning
- Despite the claim, the theoretical analysis in Section 3 does not shed insight on how global rebalancing is better than local rebalancing -- I will elaborate more on the next section
- This is also questionable whether one can estimate local label distribution from LPI - will elaborate in the next section
- It is also not clear which of the re-balance baseline corresponds to the analysis of Section 3; and more importantly, how these baselines are transplanted into the context of FL., i.e. are they applied locally or globally  -- do they all require sharing local label distribution etc.

**Summary Of The Paper:**

This paper aims to address the issue of heterogeneous, imbalanced label ratio among clients of a federated learning system. That is, the setting is that the global label distribution has a long tail; and each local dataset has imbalanced label ratio.

Previously, this can be addressed by re-balancing strategies which can be applied to either each client separately or to the server directly. Local re-balancing appear to perform worse than global re-balancing but on the flip side, global re-balancing requires local clients to share their label distribution. This does not preserve the privacy of data.

To combine the best of both worlds, the paper proposes a concept of local proxy information (LPI) that (1) makes use of some aggregated form of local parameter gradient to (roughly speaking) summarize statistics of the local label distribution; and (2) can be combined via weighted averaging in a fashion almost similar to the direct combination of local label distributions albeit with the exclusion of those with negative local proxy.

The combined proxy is referred to as the global proxy information (GPI), which is core to the proposed label-distribution ensemble learning. First, cosine distances between LPI and GPI are used to partition clients into groups. Then, in each round, an expert component is created for each group: alpha x S clients will be selected for the intended group; while (1 - alpha) x S client will be selected for the rest.

Once created, selected clients will download its corresponding expert model, bake in local data using the balanced softmax loss parameterized by the GPI and send the updated expert back. Next, to complete the communication round, the server aggregates updates for each expert; the ensemble is the average of those experts.

The proposed algorithm is tested on long-tailed versions of CIFAR-10, CIFAR-100 and ImageNet. The comparison baselines include FedAvg and many other class-prior based re-balance algorithms.

**Summary Of The Review:**

The paper presents a new idea for handling the long-tailed and imbalanced label distributions in FL. However, key concerns arise regarding (1) the lack of comparison with other types of FL algorithms that could potentially handle the same issue; (2) ambiguities over what the theoretical analysis implies; as well as potential flawed claim on privacy preservation of LPI & GPI. If these issues are addressed properly, the paper would be acceptable but for now, my preliminary assessment is that it is marginally below bar. But, I do look forward to further discussion with the authors to (hopefully) recalibrate my rating.

---

> ### Author Response · Authors · 2022-11-18
> **Response to Reviewer 5sfY Part 4/4**
>
> **Q7: How are the hyper-parameters such as M and alpha selected?**
>
> **R7**: Thanks for your comment. Two hyper-parameters M and $\alpha$ are included in our method.
>
> - For $\alpha$, we did the ablation study on the CIFAR100-LT and found the performance of our method is not sensitive to its value in the range of 0.2~0.8. The results are shown in the following table and Figure 4 in our manuscript. We experimentally tune $\alpha$ and set it to 0.4.
>
>   | Value of $\alpha$ | Accuracy |
>   | :---------------: | :------: |
>   |         0         |  38.22   |
>   |        0.2        |  40.14   |
>   |        0.4        |  40.42   |
>   |        0.6        |  39.89   |
>   |        0.8        |  38.49   |
>   |        1.0        |  36.19   |
>
> - For M, we trade off model performance and communication cost and set it to 3. We supplemented our experiments with different values of M in the following table.
>
>   | Expert Number M | Imbalance ratio = 100 | Imbalance ratio = 50 | Parameters (Million) |
>   | :-------------: | :-------------------: | :------------------: | :------------------: |
>   |        1        |         37.19         |        41.91         |         0.46         |
>   |        2        |         38.71         |        44.92         |         0.52         |
>   |        3        |         40.42         |        45.16         |         0.77         |
>   |        4        |         41.68         |        46.88         |         1.02         |
>   |        5        |         41.85         |        46.59         |         1.27         |
>
> **References mentioned in this response are as follows.**
>
> [1] Li, Tian, et al. "Federated optimization in heterogeneous networks." *Proceedings of Machine Learning and Systems* 2 (2020): 429-450.
>
> [2] Karimireddy, Sai Praneeth, et al. "Scaffold: Stochastic controlled averaging for federated learning." *International Conference on Machine Learning*. PMLR, 2020.
>
> [3] Mendieta, Matias, et al. "Local Learning Matters: Rethinking Data Heterogeneity in Federated Learning." *Proceedings of the IEEE/CVF Conference on Computer Vision and Pattern Recognition*. 2022.
>
> [4] Li, Tian, et al. "Ditto: Fair and robust federated learning through personalization." *International Conference on Machine Learning*. PMLR, 2021.
>
> [5] Collins, Liam, et al. "Exploiting shared representations for personalized federated learning." *International Conference on Machine Learning*. PMLR, 2021.
>
> [6] Ren, Jiawei, et al. "Balanced meta-softmax for long-tailed visual recognition." *Advances in neural information processing systems* 33 (2020): 4175-4186.

---

> ### Author Response · Authors · 2022-11-18
> **Response to Reviewer 5sfY Part 3/4**
>
> **Q4: For the same problem, personalized federated learning could be applied as a viable solution technique. The authors should discuss this; and provide empirical comparison to advocate for the practical significance of the proposed solution.**
>
> **R4**: Thank you for the nice suggestion. Accordingly, we add two classical personalized federated learning methods (Ditto [4] and FedRep [5]) to our experiments. The same way as training data construction is applied to partition the balanced test dataset into each client, i.e., for each class in the global dataset, we divide the samples into different clients according to the Dirichlet distribution. Then we report the averaged test accuracy.
>
> | CIFAR100-LT | Imbalance ratio = 100 | Imbalance ratio = 50 |
> | :---------: | :-------------------: | :------------------: |
> |   FedAvg    |         34.48         |        36.84         |
> |    Ditto    |         33.45         |        37.03         |
> |   FedRep    |         32.23         |        37.69         |
> |  Ditto+BSM  |         35.18         |        39.29         |
> | FedRep+BSM  |         35.04         |        39.17         |
> | LDAE(ours)  |         40.42         |        45.16         |
>
> - Some personalized federated learning methods can indeed tackle the privacy issues mentioned in **Q3**. For example, FedRep does not require the client to upload the gradients of the classifier to the server, so the server cannot infer the local label distribution.
> - However, we test the personalized FL methods Ditto [4] and FedRep [5] on the federated long-tailed problem, and as the table above shows, they do not perform well on the federated long-tailed problem, even with more training rounds. We believe the main reason is the distribution shift. Under the general setting of personalized FL without the long-tailed problem, the distribution of test data and training data are identical. However, considering the long-tailed problem, the training data is imbalanced and the test data is balanced for each client, resulting in a distribution shift. The personalized model focuses more on fitting the local training data distribution and therefore generalizes poorly to different data distributions. Compared with the personalized FL methods, non-personalized FL methods with a well-trained global model have a stronger generalization ability, so they can achieve better performance on the federated long-tailed problem.
>
> **Q5: It is also not clear which of the re-balance baseline corresponds to the analysis of Section 3; and how these baselines are transplanted into the context of FL.**
>
> **R5**: Thanks for the comment. We would like to explain it as follows:
>
> - The simplest loss re-weighting baseline (i.e., the inverse of the portion of class i is taken as the importance weight of class i) is considered in the analysis part of Section 3. More complicated loss re-weighting baselines such as LDAM, BSM and LADE, as well as other re-balance baselines are used in the following experiments with real datasets.
> - To transplant the centralized re-balance baselines into the context of FL, we applied them in local training on clients with different class priors, as shown in Table 2 in our manuscript. The detailed descriptions are shown in the following.
>   + Baselines are combined with local label distribution  (i.e., Local Re-balance part in Table 2), where they do not require sharing local label distribution among different clients.
>   + Baselines are combined with global label distribution  (i.e., Global Re-balance part in Table 2), which is shared among different clients and needs clients to upload their local label distribution to the server.
>   + Baselines are combined with global proxy information (i.e., GPI part in Table 2), which is shared among different clients but no other information is required to be uploaded by clients.
>
> **Q6: It is also strange local clients with negative LPI were excluded from the aggregation that produces GPI -- can the authors elaborate more?**
>
> **R6**: According to Eq. (6) on page 5, the negative value of LPI means it is dominated by $Z_i$, i.e., the predicted value of samples from other classes in this class, which is unstable. Besides, it provides no useful information for re-balance because it does not relate to the label distribution. The negative LPI in the GPI calculation weakens the effect of GPI, so we choose to exclude it.

---

> ### Author Response · Authors · 2022-11-18
> **Response to Reviewer 5sfY Part 2/4**
>
> **Q2: The theoretical analysis in Section 3 does not shed insight on how global rebalancing is better than local rebalancing. In my opinion, the delta term in Theorem 1 needs more intuitive interpretation.**
>
> **R2**: We greatly appreciate this highly insightful comment. After double-checking, we have deleted the absolute symbol of the delta term in Theorem 1, and the proof still holds. Please refer to Theorem 1 and Appendix B for the proof in the revised manuscript.
>
> - Since the delta value is always positive under our assumptions (see Section 3.1), the $G_{l} $ term is larger than the $G_{g} $ term, showing that the loss with the local re-balance strategy is always larger than the loss with the global re-balance strategy (the lower loss the better). This motivates us to believe that global re-balancing is better than local rebalancing.
> - We modify the interpretations of Theorem 1 in the revised manuscript: (1) The global re-balance strategy for federated long-tailed problem optimizes the same objective function as the re-balance strategy on the centralized dataset. (2) The local re-balance strategy with a larger objective is less effective than the global re-balance strategy with a smaller objective. (3) There exists a re-balance strategy derived from the global perspective, which yields a smaller objective than the local re-balance strategy.
>
>
> **Q3: This is also questionable whether one can estimate local label distribution from LPI. Regarding the privacy claim: if clients can be clustered in terms of their label distribution similarity using the LPI then it is actually not safe to say the LPI cannot be used to infer the local label distribution.**
>
> **R3**: We greatly appreciate this highly insightful comment. We would like to explain it from the following aspects:
>
> - Firstly, to the best of our knowledge, all previous re-balance methods in FL need extra information (e.g., a small balanced dataset on the server) to learn a balanced global model. However, our LDAE doesn't need it. In short, our method has the same privacy safe level as the vanilla federated learning, and has a higher privacy safe level than existing re-balance methods.
> - Secondly, we would like to clarify that the clients are clustered in terms of *the similarity of LPI*, rather than *the similarity of label distribution*. This point can be derived from the definition of LPI in Eq. (4) and Theorem 2. Specifically, LPI is defined based on the gradients and can be approximated by the difference between the local class number and the prediction probability, while the label distribution cannot be directly derived based on LPI. We are sorry for the inaccurate claim that *each expert focuses on a group of clients with similar class distributions*` in Section 3.3 in our original manuscript, and we have updated it in the revised version.
> - Moreover, if the server holds a balanced validation set, we highly agree with the reviewer that the local class distribution could be approximated by solving a linear system based on the stable $\mathcal{Z}_i$ and $s_j$ terms in Eq. (5) once the training converged. Thanks for this smart suggestion. It is really a privacy threat to both vanilla FL and our method. One possible solution is adding random perturbation on the updated gradients, as did in differential privacy. We plan to explore this idea and more advanced strategies to enhance the privacy preservation of our method. Your suggestion about using the personalized FL methods is also a capable solution to this privacy problem. We have added some experiments and discussions. Please see the following **R4**.
> - *In summary*, our method using LPI doesn't require the local label distribution, but we didn't provide more protection than the vanilla FL. Although the local label distribution cannot be directly derived from LPI (*i.e.*, the gradients), it may be approximated by other smart strategies (like the reviewer's suggested method). Thus, we have clearly illustrated the exact meaning of *privacy preservation* in the revised manuscript, to avoid any over-claim or confusion. We appreciate again this insightful comment, which inspires us to develop a more advanced method with higher privacy preservation.

---

> ### Author Response · Authors · 2022-11-18
> **Response to Reviewer 5sfY Part 1/4**
>
> Dear Reviewer 5sfY,
>
> We sincerely appreciate your precious time and constructive comments, and are greatly encouraged by your high recognition about our efforts in addressing the issue of heterogeneous, imbalanced label ratio among clients of a federated learning system. The code is available at https://anonymous.4open.science/r/anonymous-001/.
> In the following, we would like to answer your concerns separately.
>
> ---
>
> **Q1: Lacking coverage of other types of FL algorithms that deal with data heterogeneities, such as personalized federated learning.**
>
> **R1**: Thanks for this constructive comment. According to your suggestion, we add new experiments with other types of FL algorithms that deal with data heterogeneity concerning non-personalized FL and personalized FL, respectively.
>
> - FedProx [1], SCAFFOLD [2], and FedAlign [3] as baselines on the CIFAR100-LT are recent non-personalized FL algorithms for dealing with data heterogeneity, i.e., FedProx and SCAFFOLD use different proximal terms. FedAlign designs a regularization method for local training. Please see the following table.
>
>   |   CIFAR100-LT    | Imbalance ratio = 100 | Imbalance ratio = 50 |
>   | :--------------: | :-------------------: | :------------------: |
>   |      FedAvg      |         34.48         |        36.84         |
>   |     FedProx      |         34.16         |        38.77         |
>   |     SCAFFOLD     |         34.70         |        40.44         |
>   |     FedAlign     |         35.36         |        39.80         |
>   |  FedAvg+BSM+GPI  |         37.19         |        41.91         |
>   | FedProx+BSM+GPI  |         37.78         |        42.33         |
>   | SCAFFOLD+BSM+GPI |         38.55         |        42.27         |
>   | FedAlign+BSM+GPI |         39.21         |        43.66         |
>   |    LDAE(ours)    |         40.42         |        45.16         |
>
>   + As shown in the table, the non-personalized FL methods (FedProx, SCAFFOLD, FedAlign) are effective in FL with relatively mild class imbalance (imbalance ratio = 50), but they are less effective when class imbalance is severe (imbalance ratio = 100).
>   + More importantly, when combined with our re-balance strategy (BSM [6] with GPI), these methods can obtain a significant performance improvement for various imbalance ratios.
>
>
>  - Ditto [4] and FedRep [5] focus on personalized FL settings. We compare them with our method on the CIFAR100-LT. Please see the following table.
>
>    | CIFAR100-LT | Imbalance ratio = 100 | Imbalance ratio = 50 |
>    | :---------: | :-------------------: | :------------------: |
>    |   FedAvg    |         34.48         |        36.84         |
>    |    Ditto    |         33.45         |        37.03         |
>    |   FedRep    |         32.23         |        37.69         |
>    |  Ditto+BSM  |         35.18         |        39.29         |
>    | FedRep+BSM  |         35.04         |        39.17         |
>    | LDAE (ours) |         40.42         |        45.16         |
>
>    - To test the effect of the personalized FL methods Ditto [4] and FedRep [5] on the long-tailed FL problem, we use the same way as training data construction to partition the balanced test dataset into each client. That is to say, for each class in the global dataset, we divide the samples into different clients according to the Dirichlet distribution. Then we report the averaged test accuracy.
>    - From the above table, we can see that the personalized FL approaches perform poorly on the federated long-tailed problem, even with more training rounds. We believe the main reason is the distribution shift. Under the general setting of personalized FL without the long-tailed problem, the distribution of test data and training data are identical. However, considering the long-tailed problem, the training data is imbalanced and the test data is balanced for each client, resulting in a distribution shift. The personalized model focuses more on fitting the local training data distribution and therefore generalizes poorly to different data distributions. Compared with the personalized FL methods, non-personalized FL methods with a well-trained global model have a stronger generalization ability, so they can achieve better performance on the federated long-tailed problem.

---

> ### Author Response · Authors · 2022-11-22
> **Have our responses addressed your concerns?**
>
> Thank you again for your careful read and detailed comments about our manuscript. We were just wondering whether you could kindly let us know whether you are satisfied with our responses, or whether you have any remaining questions which we might be able to address.

---

### Official Review · Reviewer_EiQm · 2022-10-25

**Confidence:** 3
**Correctness:** 3
**Technical Novelty And Significance:** 3
**Empirical Novelty And Significance:** 3
**Recommendation:** 6

**Clarity, Quality, Novelty And Reproducibility:**

Most parts of the paper are clear, however, the presentation of Local Proxy Information and global Proxy Information seems not very clear. Can you give an illustrated figure for more intuitive understanding?

My biggest concern still lies in the setting of the paper. Is the long-tailed setting in the paper reasonable? do we have other options? Happy to discuss more about this.

**Strength And Weaknesses:**

Strength:

1. Overall, the paper is clear and easy to follow.

2. They especially consider the importance of privacy in federated learning, while alleviating the problem of the long tail.

3. The analysis of the proposed method is comprehensive and the results of the experiment are also good.

Weaknesses:

1. The setting of this paper is not clear. In "dataset and setup" in Section 4, a brief operation of how to simulate FL is introduced. "the global dataset is partitioned into 100 local datasets with the Dirichlet distribution". For each local dataset, do they draw samples from each class by following Dirichlet distribution?

2. The combination of federated learning and long-tailed datasets is good. However, in this paper, only a primitive federated method is considered. Most of the baselines focus on different reweighting methods in long-tailed recognition. I am curious about the results of directly applying the latest federate methods into long-tailed datasets, including the works discussed in the related works.

3. Can you show some visualization examples of the data distribution of local clients?

**Summary Of The Paper:**

This paper proposes a new method to deal with long-tailed recognition in a federated setting. Different from previous works, they pay much attention to privacy, which is the core idea of federated learning. Since the global label distribution is inaccessible in this setting, they instead use the model updates of the clients as surrogates. In the experiments, they show the proposed method achieves comparable results, compared to methods with global priors.

**Summary Of The Review:**

The paper considers a more strict long-tailed recognition setting by retaining the constraint of privacy in federated learning. This paper is well presented and the results are also good. However, some points are worth further discussing.

---

> ### Author Response · Authors · 2022-11-18
> **Response to Reviewer EiQm Part 2/2**
>
> **Q4: I am curious about the results of directly applying the latest federate methods into long-tailed datasets, including the works discussed in the related works.**
>
> **R4**: Thank you for the nice suggestion. Accordingly, we add new experiments with FedProx [7], SCAFFOLD [8] and FedAlign [9] as baselines on the CIFAR100-LT. They are recent FL algorithms for dealing with data heterogeneity. More precisely, FedProx and SCAFFOLD design different proximal terms and FedAlign designs a regularization method for local training. The results are shown in the following table.
>
> |   CIFAR100-LT    | Imbalance ratio = 100 | Imbalance ratio = 50 |
> | :--------------: | :-------------------: | :------------------: |
> |      FedAvg      |         34.48         |        36.84         |
> |     FedProx      |         34.16         |        38.77         |
> |     SCAFFOLD     |         34.70         |        40.44         |
> |     FedAlign     |         35.36         |        39.80         |
> |  FedAvg+BSM+GPI  |         37.19         |        41.91         |
> | FedProx+BSM+GPI  |         37.78         |        42.33         |
> | SCAFFOLD+BSM+GPI |         38.55         |        42.27         |
> | FedAlign+BSM+GPI |         39.21         |        43.66         |
> |   LDAE (ours)    |         40.42         |        45.16         |
>
> According to the above table, we have the following analysis.
>
> - The FL methods (i.e., FedProx, SCAFFOLD, FedAlign) are effective in FL with relatively mild class imbalance (imbalance ratio = 50), but they are less effective when class imbalance is severe (imbalance ratio = 100).
> - More importantly, when combined with our re-balance strategy (BSM [10] with GPI), these methods can obtain a significant performance improvement in various imbalance ratios.
> - Comparably, our method LDAE achieves the best performance with two different imbalance ratios.
>
> **Q5: The presentation of Local Proxy Information and global Proxy Information seems not very clear. Can you give an illustrated figure for more intuitive understanding?**
>
> **R5**: According to your suggestion, we provide an illustrated figure to show the LPI and GPI calculation process. Please see Figure 6 in Appendix F.
>
> - At the first round, the server receives the model updates uploaded by each client. For client $k$, the LPI of class 1 is calculated by summing the updated values of the weights connected to neuron $x^1$ according to Eq. (2).
> - Then, we obtain the GPI of class 1 by calculating the weighted sum of the LPI. The GPI of other classes is calculated in the same way.
>
> **References mentioned in this response are as follows.**
>
> [1] Yurochkin, Mikhail, et al. "Bayesian nonparametric federated learning of neural networks." *International Conference on Machine Learning*. PMLR, 2019.
>
> [2] Li, Qinbin, Bingsheng He, and Dawn Song. "Model-contrastive federated learning." *Proceedings of the IEEE/CVF Conference on Computer Vision and Pattern Recognition*. 2021.
>
> [3] He, Chaoyang, et al. "Fedml: A research library and benchmark for federated machine learning." *arXiv preprint arXiv:2007.13518* (2020).
>
> [4] Wang, Lixu, et al. "Addressing class imbalance in federated learning." *Proceedings of the AAAI Conference on Artificial Intelligence*. Vol. 35. No. 11. 2021.
>
> [5] Shen, Zebang, et al. "An Agnostic Approach to Federated Learning with Class Imbalance." *International Conference on Learning Representations*. 2021.
>
> [6] Shang, Xinyi, et al. "Federated Learning on Heterogeneous and Long-Tailed Data via Classifier Re-Training with Federated Features." *arXiv preprint arXiv:2204.13399* (2022).
>
> [7] Li, Tian, et al. "Federated optimization in heterogeneous networks." *Proceedings of Machine Learning and Systems* 2 (2020): 429-450.
>
> [8] Karimireddy, Sai Praneeth, et al. "Scaffold: Stochastic controlled averaging for federated learning." *International Conference on Machine Learning*. PMLR, 2020.
>
> [9] Mendieta, Matias, et al. "Local Learning Matters: Rethinking Data Heterogeneity in Federated Learning." *Proceedings of the IEEE/CVF Conference on Computer Vision and Pattern Recognition*. 2022.
>
> [10] Ren, Jiawei, et al. "Balanced meta-softmax for long-tailed visual recognition." *Advances in neural information processing systems* 33 (2020): 4175-4186.

---

> > ### Comment · Reviewer_EiQm · 2022-11-25
> > **Response to rebuttal**
> >
> > The feedback in the rebuttal has well-addressed my concerns.
> >
> > As discussed above, my concerns in the setting is partially addressed.
> >
> > Happy to see more papers related to study this topic.
> >
> > Finally, I raise my score to 6.

---

> > > ### Author Response · Authors · 2022-11-25
> > > **Response to Reviewer EiQm**
> > >
> > > Dear Reviewer EiQm,
> > >
> > > We are highly encouraged by your positive comments. We will improve the final version of the manuscript according to the constructive and helpful discussions with you.
> > >
> > > Best Regards,
> > > The Authors

---

> ### Author Response · Authors · 2022-11-18
> **Response to Reviewer EiQm Part 1/2**
>
> Dear Reviewer EiQm,
>
> Thank you for your valuable time and thoughtful comments. Below, we address all the points you raised. We have added figures to the appendix to answer your questions about data heterogeneity simulation and how the proxy information is calculated.
>
> ---
>
> **Q1: The setting of this paper is not clear. For each local dataset, do they draw samples from each class by following the Dirichlet distribution?**
>
> **R1**: Thanks for this constructive comment. We clarify our data setting as follows:
>
> - Step 1: generating $\mathbf{p_c} \sim Dir_K(\alpha_{dir})$ for class $c$ and normalizing it.
> - Step 2: allocating the $p_{c,k}$ proportion of the samples in class $c$ to client $k$.
>
> The setting simulates the non-iid data distribution of the training dataset, which is commonly considered in FL [1] [2] [3]. The above setting has been added in the "Datasets and Setup" of the revised manuscript (highlighted in blue).
>
>
> **Q2: Can you show some visualization examples of the data distribution of local clients?**
>
> **R2**: Thanks for this suggestion. We add some visualization examples in Figure 5 in Appendix E, which includes different data heterogeneity degrees.
>
> - The subfigures (i.e., d, e, f) show the data distributions of local clients with different values of $\alpha_{dir}$, i.e., the parameter of the Dirichlet distribution, and we also show the corresponding probability density of the Dirichlet distribution in subfigures a, b and c.
> - The illustration of the global label distribution is included in subfigure g, which is a long-tailed distribution.
> - The figure shows that: 1) each local dataset follows an imbalanced distribution; and 2) the distributions among local datasets are different. The constructed distributions exactly satisfy the considered setting in this work, *i.e.*, heterogeneous imbalance (please see Definition 1 in the manuscript).
>
> **Q3: My biggest concern still lies in the setting of the paper. Is the long-tailed setting in the paper reasonable? do we have other options? Happy to discuss more about this.**
>
> **R3**: We greatly appreciate this highly insightful comment. We would like to discuss several different possible settings for the federated long-tailed problem from the following three perspectives:
>
> - Global long-tailed or local long-tailed. The long-tailed problem can exist in different situations:
>   + The global data distribution is long-tailed, and the data distributions of local clients are not assumed in advance. This is the situation considered in this work. This setting is more general and challenging because large-scale data in real-world applications is often naturally long-tailed.
>   + The local dataset of each client follows individual long-tailed distribution, while the global data distribution is not assumed in advance.
>     For cross-device FL considered in our work (client number is 100), as the samples on the clients are usually relatively small and, due to heterogeneity, some clients may only have a small number of classes, it is reluctant to assume that the data on the clients follow a long-tailed distribution. Instead, perhaps in cross-silo FL, where there are only a few clients (such as 5 and 10) and the sample number on each client is pretty large, one could consider the setting that the data on the local client follows a long-tailed distribution.
> - Homogenous or heterogenous. When considering that the global dataset is long-tailed, the label distribution of local datasets on clients may be the same (homogenous) or different (heterogeneous). The latter one is usually considered by the literature since it is more realistic and challenging.
> - Given the fact that we consider heterogeneous distribution among different local datasets, we discuss two approaches to simulate the heterogeneity.
>   + Among the papers that solve a similar problem to ours, [4] first pre-selects a specified number of classes and then samples data from these classes for each client.
>   + [5] and [6] draw the samples based on the Dirichlet distribution.
>   + Both of them are common settings in FL. However, we believe that the second one is more suitable, as the class distribution of clients obtained this way is more diverse, and the variable class number of clients is more realistic.

---

> ### Author Response · Authors · 2022-11-22
> **Have our responses addressed your concerns?**
>
> Thank you again for your careful read and detailed comments about our manuscript. We were just wondering whether you could kindly let us know whether you are satisfied with our responses, or whether you have any remaining questions which we might be able to address.

---

### Decision · Program_Chairs · 2023-01-20

**Decision:**

Reject

**Justification For Why Not Higher Score:**

NA

**Justification For Why Not Lower Score:**

NA

**Metareview: Summary, Strengths And Weaknesses:**

The paper proposes a Label-Distribution-Agnostic Ensemble (LDAE) for Federated learning with heterogeneous or long tailed distribution data. The proposal involves using multiple experts and their control their grouping based on a novel global objective with privacy constraints. Experiments showed significant gains.

The paper claims about privacy and the settings are not well formed and requires additional scrutiny. Specially, the capability to do group users based on similar data distributions and other assumptions made need more thorough evaluations. Given that privacy preserving is one of the main objectives, this problem becomes critical as pointed out by one of the reviewers after significant discussions.

**Summary Of Ac-Reviewer Meeting:**

The reviewers and authors have a very detailed discussions and clearly the privacy issue seems critical here.
There needs to be additional validation and verification of privacy claims in real practical settings before the manuscript gets published.

Also, reviewers did point out that the paper should compare and contrast with several existing ideas around grouping devices with similar distributions to convincingly demonstrate the value of the ideas presented.